# Atomic scale volume and grain boundary diffusion elucidated by in situ STEM

Peter Schweizer [1] ✉, Amit Sharma[1], Laszlo Pethö[1], Emese Huszar[1], Lilian Maria Vogl[1], Johann Michler[1,2] & Xavier Maeder[1]

Diffusion is one of the most important phenomena studied in science ranging from physics to biology and, in abstract form, even in social sciences. In the field of materials science, diffusion in crystalline solids is of particular interest as it plays a pivotal role in materials synthesis, processing and applications. While this subject has been studied extensively for a long time there are still some fundamental knowledge gaps to be filled. In particular, atomic scale observations of thermally stimulated volume diffusion and its mechanisms are still lacking. In addition, the mechanisms and kinetics of diffusion along defects such as grain boundaries are not yet fully understood. In this work we show volume diffusion processes of tungsten atoms in a metal matrix on the atomic scale. Using in situ high resolution scanning transmission electron microscopy we are able to follow the random movement of single atoms within a lattice at elevated temperatures. The direct observation allows us to confirm random walk processes, quantify diffusion kinetics and distinctly separate diffusion in the volume from diffusion along defects. This work solidifies and refines our knowledge of the broadly essential mechanism of volume diffusion.

In a crystalline lattice diffusion can be understood as a hopping process of atoms driven by thermal vibrations[1]. Atoms can hop from lattice site to lattice site via a vacancy-mediated process[2–4] or move through the lattice by making use of interstitial sites[4,5]. Which mechanism is dominant depends on the type of lattice and elements involved with interstitial diffusion being frequently observed for light elements[6] and vacancy assisted diffusion being the main mechanism for self-diffusion[7,8] and diffusion of heavier impurity elements[9]. Several more complex mechanisms have been put forward like ring diffusion[10] or crowdion migration[11] but clear experimental proof for these mechanisms is lacking. In general, volume diffusion is described by an Arrhenius relationship in which the coefficient of diffusion depends on an activation energy and the temperature. However, since the early days of volume diffusion experiments, a deviation from a simple Arrhenius relationship especially at low temperatures has been observed[7,12]. In many cases, low-temperature data is neglected altogether, attributing it to fast diffusion along defects, especially grain boundaries[13]. However, all research regarding volume diffusion relies on indirect measurements using for instance radioactive tracer profiles[14] or characteristic diffraction spots[15], limiting their conclusions on diffusive mechanisms. Instead, here we use in situ high resolution scanning transmission electron microscopy (STEM) to follow the diffusion of single atoms directly. To do this we combine a heavy impurity element (W) with a light matrix element (Al,Cu) to obtain the necessary contrast. This allows us to differentiate between pure volume and grain boundary diffusion and make assertions towards diffusive mechanisms. Furthermore, low temperature diffusion is readily accessible due to the ability to detect even single atomic hops, the smallest possible displacement in a lattice.

## Results and discussion

Figure 1a schematically shows diffusive random walks in a volume for substitutional impurities in a face-centred cubic (FCC) lattice. To experimentally see single atoms in such a volume, a combination of a light matrix and heavy impurity element is highly beneficial to obtain a good contrast in high-angle annular dark-field STEM. The principle

[1]Swiss Federal Laboratories for Materials Science and Engineering (Empa), Laboratory for Mechanics of Materials and Nanostructures, Feuerwerkerstrasse 39, 3602 Thun, Switzerland. [2]École Polytechnique Fédérale de Lausanne (EPFL), 1015 Lausanne, Switzerland. ✉e-mail: peter.ps.schweizer@outlook.com

**Fig. 1 | Volume diffusion of impurity atoms in a metal lattice. a** 3D rendering of random walks of impurity atoms inside an face-centred cubic crystal. **b** STEM image of single impurity atoms in an Al film at room temperature (RT). **c** Experimentally observed random walks of impurity atoms in copper at 385 °C over a time of 120 s. The arrows signify the movement of an atom along the positions highlighted in red during the timeframe. The raw images of the sequence and the full animation are shown in Supplementary Fig. 4 and Supplementary Movie 1.

feasibility of this approach has been demonstrated earlier by Ishikawa et al. who looked at impurity atoms in a semiconductor crystal[16]. In this work, we incorporated dilute tungsten impurities (around 150 ppm) in a copper or aluminium thin film (see Supplementary Fig. 1 for an overview of the deposited films) using a physical vapour deposition process. Fig. 1b) shows a STEM image of impurity atoms in a metal thin film demonstrating the feasibility of the detection of these atoms (see Supplementary Fig. 2 for a comparison between experimental and simulated images for both Cu and Al as well as Supplementary Fig. 3 for simulations of impurity atoms at different positions in the lattice). Figure 1c shows experimental diffusive paths of tungsten atoms in a copper matrix at 385 °C over a time period of 120 s. The pathways of the impurities have been highlighted in red (original image and snapshots of the image series in Supplementary Fig. 4 and movie in Supplementary Movie 1). Additional contrast potentially stemming from atoms on a surface position or grain boundary site is visible in the bottom left (not highlighted). The atoms indeed perform random walks through the lattice by jumping from lattice site to lattice site. Different impurity atoms move at different apparent speeds, meaning they perform a different number of jumps throughout the given timeframe, with some atoms being entirely stationary during the relatively short time of observation. In some cases, atoms suddenly appear (see Supplementary Movie 1, top left) or disappear in the crystal which may be attributed to jumps from the surface into the volume or vice versa. To make sure that diffusion is thermally stimulated, beam-off experiments have been performed where impurities were observed, then the electron beam was switched off. The sample was then heated for a defined time interval, after which the beam was switched on and the impurities were located again. These experiments yielded similar movements for the impurities with the drawback that individual random walks cannot be traced (see Supplementary Fig. 5 for a beam-off experiment). Looking at a single diffusion path more closely (shown in Fig. 2 and Supplementary Movie 2) we can see all the characteristics of a random walk: the atom performs jumps at random time intervals in random directions. In the given example, the atom moves away from the origin up to about 70 s, stagnates at a relatively constant distance until 140 s and then comes closer to the origin again until the end of the series. The random walk appears to be largely uncorrelated, hinting at many independent encounters with vacancies[17]. In the shown example, the copper crystal is oriented in a <100> zone axis which means that there are two possible projected jump distances. In the FCC lattice there are 12 nearest neighbour positions in <110> type directions that an atom can jump to by a vacancy-mediated process. Four nearest neighbour jumps are fully contained in the plane of observation giving an observable jump length of 2.55 Å. The other eight possible jumps are partially projected into the plane of observation resulting in an apparent hopping distance of 1.8 Å. Counting the number of jumps in the different directions it is possible to

infer the relative in-plane and out-of-plane movements of the impurities. In our experiments, they follow the 4:8 ratio expected for a random walk in which all nearest neighbour jumps have the same probability. Together with statistics about the directionality of in-plane movement, we can conclude that the atoms move isotropic within the statistical limits of our observation (see Supplementary Fig. 6). The data is consistent with volume diffusion and not with surface diffusion for which there are only four nearest neighbour positions on the 100 surface available to jump to assuming a hopping mechanism[18].

While each individual atom can return to its origin during observation, on average, atoms move away from their origin over time following a square root relationship[19].

$$r = \sqrt{2nDt} \qquad (1)$$

Where $r$ is the distance from the origin, $n$ is the number of dimensions of the random walk, $D$ is the coefficient of diffusion and $t$ is time. In this case $n = 2$ due the observation being a 2D projection of a 3D random walk. As previously mentioned, we see an isotropic diffusion ($D_x = D_y = D_z$) which is commonly expected for cubic metals in the absence of a chemical gradient or other driving forces. Based on our experimental approach, we can now directly access the coefficient of diffusion for volume diffusion of tungsten impurities in a metal matrix in the temperature range of 250–475 °C (the quantification results are discussed further down below). At 250 °C only single atomic jumps can be observed over periods of ≥10 min giving us a lower limit for the temperature at which diffusion can be quantified using this method. For comparison, when looking at surface diffusion, even at room temperature a considerable amount of movement would be expected as the activation energies are typically much lower than in the volume[20]. For instance surface self-diffusion of tungsten has an activation energy between 0.5 and 0.9 eV[21] which would lead to coefficients of diffusion in the range of $10^{-19}$ m²/s at room temperature. Such values would indicate a mean displacement in the range of several Angstroms per second without any additional heating. Indeed, we are able to observe this for atoms on the exposed part of the TEM grids, which show room temperature surface diffusion and cluster formation characteristic for that process (see Supplementary Fig. 7). At elevated temperatures, surface diffusion becomes too fast to be tracked with mean displacements of hundreds of nm per second at a temperature of 200 °C. In comparison, in the volume only at temperatures higher than 475 °C the diffusive movement becomes too rapid to be tracked (for illustration diffusion at 450 °C is shown in Supplementary Movie 3). In addition, electron beam knock-on effects for copper have to be considered in this temperature range as well[22] whereas this is not an issue at lower temperatures (8.2 eV maximum energy transfer compared to threshold displacement energy of 19 eV

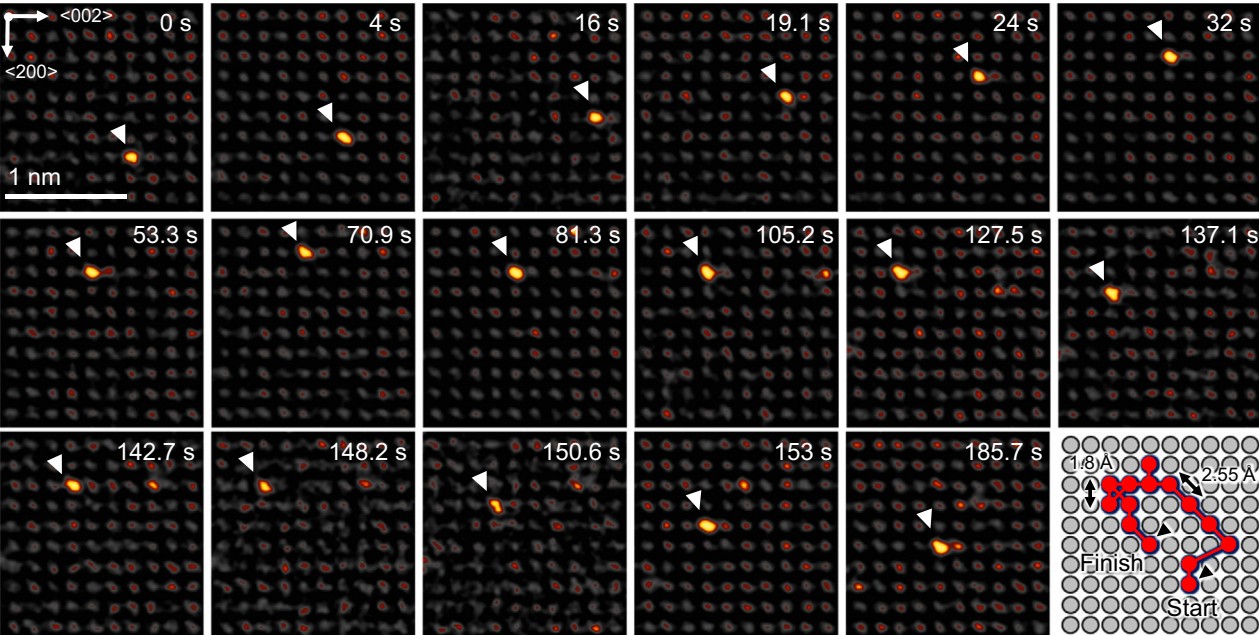

**Fig. 2 | Image series of a single atom random walk at 385 °C.** The copper crystal orientation is 100. The tungsten atom jumps from lattice site to lattice site in a random walk, with different time periods passing in between jumps. Throughout the trajectory, the atom first moves further away and then later closer towards its origin. The full trajectory is shown in Supplementary Movie 2. For the calculation of the diffusion coefficient the distance between start and finish of the random walk and the total time of observation is taken.

at room temperature[23]). The effect of knock-on on diffusion can be readily observed when using aluminium as a matrix material instead of copper. 200 keV electrons have enough energy to displace Al atoms from their lattice positions at room temperature (maximum energy transfer of 19.5 eV compared to the threshold displacement energy of 16 eV[23]), either ejecting them from the solid or forcing them into interstitial sites, leaving behind vacancies[24]. This leads to an accelerated diffusion of impurities starting already at room temperature and an additional mechanism of interstitial-type diffusion (see Supplementary Fig. 8 and Supplementary Movies 4 and 5).

The direct observation of diffusive trajectories enables us also to differentiate between true volume diffusion and diffusion along grain boundaries in a way not possible with indirect measurements. A type of boundary that is very common in copper are coherent twin boundaries (CTB). Although they have a much smaller energy than other types of grain boundaries[25], they are still expected to influence diffusion[26]. Fig. 3a, c) shows an exemplary diffusive motion of an impurity atom at a twin boundary (full series can be found in Supplementary Movie 6). The impurity performs a random walk however, the frequency of jumps is slightly enhanced when the atom is located at an atomic column directly at or neighbouring the boundary. In addition, there is a preference for the impurity to return to an atomic column located at the CTB. The abundance of twin boundaries in copper enables the direct quantification of diffusion in dependence of temperature for atoms at such a boundary. In contrast to CTBs, large angle grain boundaries have a much higher energy and are expected to strongly enhance diffusion. However, this is not always the case, as diluted impurities may segregate at specific boundaries[27], which significantly slows down the diffusive process instead of enhancing it. In our experiments, we were able to observe both behaviours (see Supplementary Fig. 9 for GB segregation). An example of enhanced diffusion at a grain boundary is shown in Fig. 3b, d) as well as Supplementary Movie 7. Even at 250 °C, a temperature at which only single atomic jumps were observed for pure volume diffusion, an impurity atom moves rapidly along a grain boundary. In this instance the lower grain is oriented in a 110 zone axis with the 200 plane bordering the grain boundary. The exact

orientation of the upper grain is not known, however the lattice fringes parallel to the grain boundary are indicative of a 111 plane. Throughout the entire diffusive motion, the atom stays at the grain boundary. Due to the projection in TEM and the 2D nature of grain boundary diffusion, the movement in this case appears to be one dimensional. The mechanism of diffusion along this grain boundary is a hopping process from lattice site to lattice which are also grain boundary sites. In other grain boundaries we also saw a diffusing atom making use of the excess volume to jump to grain boundary interstitial sites (see Supplementary Fig. 10). This confirms that the mechanism of grain boundary diffusion can rely on both, lattice sites and excess volume depending on the type of boundary.

By tracking the diffusive motion of many atoms, fundamental diffusion coefficients can be extracted for different temperatures. These coefficients of diffusion (D) are expected to follow an Arrhenius-type relationship:

$$D = D_0 e^{\frac{-Q}{k_B T}} \qquad (2)$$

With the maximal diffusion $D_0$, the activation energy $Q$, Boltzmann constant $k_B$ and temperature $T$. In this study, diffusion coefficients for volume diffusion and diffusion along CTBs were extracted in the temperature range of 250 °C to 475 °C. The resulting Arrhenius plots are shown in Fig. 4. For the higher temperature regime, a straight line can be fitted yielding an activation energy of 2.19 eV for pure volume diffusion in copper. This value is in line with literature data concerning metal impurities in copper (e.g. Cr: 2.3 eV, Au: 1.95 eV, Fe: 2.2 eV)[28] and is in the range of self-diffusion (2.04–2.19 eV)[29]. Interestingly, the slope of the low temperature part of the graph deviates from the high-temperature part. This behaviour has been seen in many studies on volume diffusion, however, interpretations vary. In many early studies, enhancement by grain boundary diffusion has been identified as a potential cause[13]. However, in this case we can definitively rule out this effect as only pure volume diffusion events have been used for the quantification. Later interpretations of low-temperature diffusion proposing divacancy mechanisms in addition

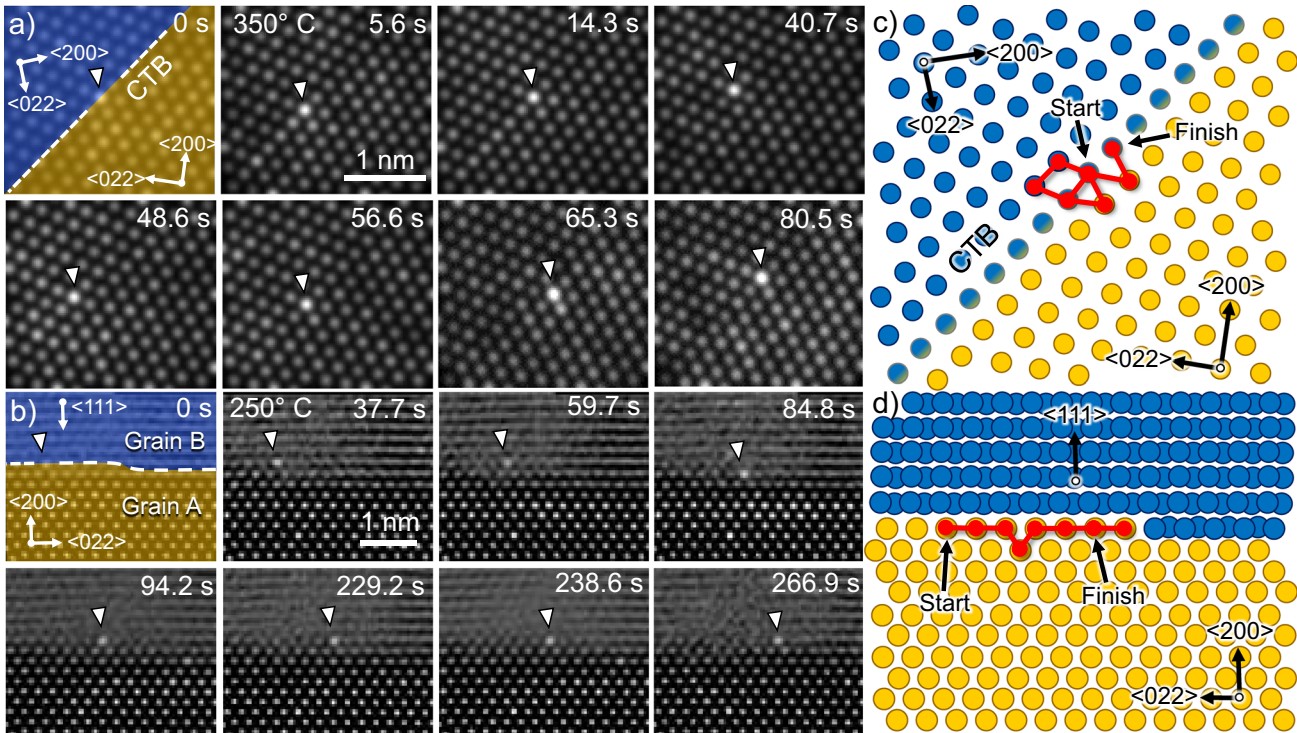

**Fig. 3 | Diffusion along grain boundaries in copper. a** Diffusion along a coherent twin boundary (CTB) at 350 °C for ca. 80 s. The number of jumps the atom performs is enhanced close to the grain boundary. In addition, it has a tendency to return to the boundary. **b** Diffusion along a high-angle grain boundary at 250 °C for ca 270 s. The atom hops directly from grain boundary site to grain boundary site.

The full trajectories can be found in Supplementary Movies 6 and 7. **c, d** Schematic representations of grain boundary structures shown in (**a**) and (**b**) with atomic columns belonging to either grain marked in blue and yellow. The total path of the diffusing atoms is shown in red.

to single vacancy mechanisms to contribute to diffusion seem more plausible[1]. However, in our experimental temperature range this contribution is still expected to be small. The lower temperature part of the Arrhenius plot yields an activation energy of 1.14 eV, which is in the range of the pure vacancy migration energy (around 1 eV[1]). It has to be noted that the material deposition by physical vapour deposition can introduce a higher than equilibrium vacancy concentration[30], which may lead to an increased mobility at lower temperatures. However, the exact effect of this remains hard to quantify. While it is unclear at this point, what exactly causes the comparably high low-temperature diffusion, we can show that it proceeds via atomic hopping processes and rule out other mechanisms such as interstitial processes, crowdion migration or an effect of grain boundaries. Comparatively, the quantification of W diffusion in Al gives us an overall activation energy as low as 0.3 eV (see Supplementary Fig. 11) which is much lower than expected and can be attributed to electron beam knock-on damage.

Due to the random walk nature of diffusion, there is a distribution of apparent 'speeds' (meaning the total distance travelled during the time of observation) of the atoms. The probability density distribution of apparent diffusion speeds at 350 °C is shown in the inset in Fig. 4a. Interestingly, this plot can be fitted using a 2D Maxwell-Boltzmann type distribution function, which gives us an equivalent mass for an ideal gas (in this case around $4.6 \times 10^3$ kg) that has the same diffusive characteristics as the impurities in the crystal. While the process of volume diffusion is fundamentally different than that of a gas, it is intriguing to see that they can be both described in the way which may be of value for modelling the process.

Uniquely for our method we are also able to extract diffusion coefficients for coherent twin boundaries. The graph follows the same general trend as the one for pure volume diffusion. Using the 'high temperature' part of the curve we find an activation energy of 1.8 eV which is lower than the volume counterpart. This is expected, as grain

boundaries, even low energy CTBs, generally enhance diffusion and there is a direct link between grain boundary diffusivity and the grain boundary energy. After the modified relation of refs. 31,32 we can relate the grain boundary energy ($E_{gb}$) to the fraction of the diffusion coefficients ($D_{gb}$ and $D$) and the mean distance of atoms in the grain boundary ($a$):

$$E_{gb} = \frac{k_B T}{2a^2} \ln\left(\frac{D_{gb}}{D}\right) \qquad (3)$$

Using our data, we can estimate a CTB Energy of approximately $E_{gb} = 31 \frac{mJ}{m^2}$ at 300 °C, which is close to expected values[33,34]. We have to stress here that this analysis is highly simplified and may only be valid for specific boundaries such as CTBs.

In summary, we demonstrated the atomic scale observation of volume diffusion of dilute tungsten atoms in an FCC metal matrix in a range of temperatures. We could experimentally confirm that atoms perform isotropic random walks through the lattice, using nearest neighbour jumps. The movement of the atoms was tracked to extract the kinetics of diffusion. Here, we were able to unambiguously separate true volume diffusion from diffusion at twin boundaries, giving us access to the activation energy of diffusion along CTB. Additionally, we were able to show that large angle grain boundaries, can either enhance diffusion or lead to the segregation of dilute impurities. The mechanism for movement along grain boundaries can be either hopping from lattice site to lattice site or the use of interstitials sites in the excess volume of the grain boundary. Finally, for low-temperature diffusion we could show that diffusion is still facilitated via vacancy-mediated nearest neighbour jumps and rule out other mechanisms. We believe these observations are transferrable to similar systems of FCC metals with metal impurities. This work can not only serve as a 'textbook' example to demonstrate the mechanism of volume

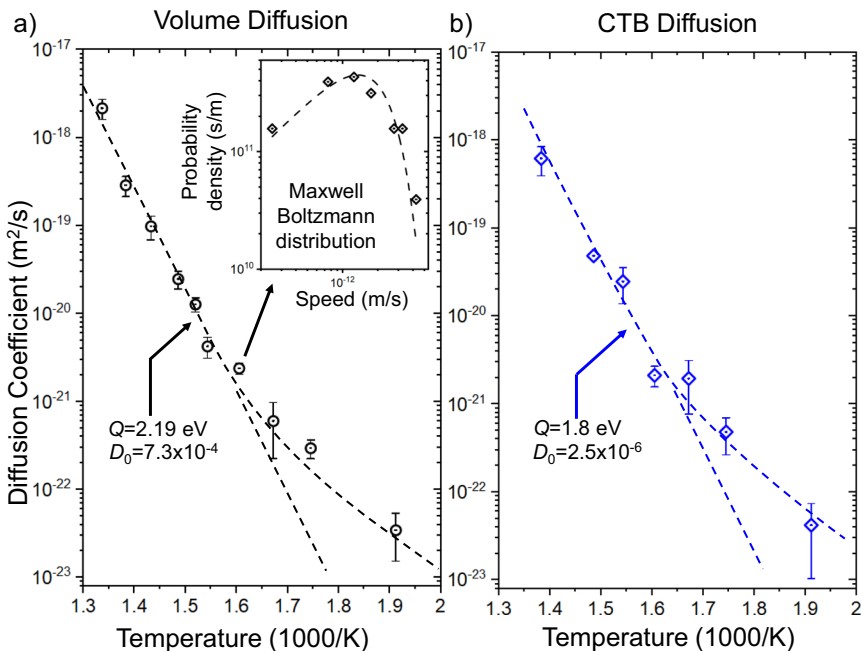

**Fig. 4 | Arrhenius plots of volume and twin boundary diffusion.** Quantification of diffusion coefficients in dependence of temperature for (**a**) volume diffusion and (**b**) coherent twin boundary diffusion. Error bars represent the standard error. Inset in (**a**) shows the distribution function of apparent speeds of atoms diffusing at 350 °C and a 2D Maxwell-Boltzmann fit to that function. Dashed lines indicate a linear fit to the high temperature regime and the low temperature deviation from that straight line. Source data are provided as a Source Data file.

diffusion in a crystal lattice but also provide insights into diffusion at defects and at low temperatures.

## Methods

### Film deposition

Thin films were deposited using co-sputtering in a Mantis QPREP 500 PVD deposition instrument. The Host film was deposited using DC Magnetron sputtering while the solutes were deposited simultaneously using a nanoparticle gun (Nanogen 50, Mantis)[35]. The nanoparticle gun uses DC sputtering and terminated gas condensation to form nanoparticles. This system can produce nanoparticles in the range of around 2–5 nm accompanied by dilute atoms which, during co-sputtering leads to finely dispersed impurities. In the here-presented experiments, the concentration of impurities was determined to be in the range of 150 ppm.

### In situ TEM heating

In situ heating was performed using a DENS Solutions Wildfire S3 heating holder. TEM was performed in a probe-corrected Titan Themis 200 G3 at 200 kV. The probe current used in the measurements was 200 pA with typical dwell times between 0.5–2 μs (final frame times between 0.5 and 2 s) and a scan size of 1024 × 1024 pixels. An additional negligible heating effect due to the electron beam was estimated to be around 0.03 K by the method presented by Egerton[36]. The dose rate was varied between $10^4$ and $10^5$ e⁻/A²/s which did not influence the diffusion of W in Cu but accelerated diffusion of W in Al.

### Data evaluation

Subpixel accurate image alignment was performed using 2D Gaussian fitting of the peak of the cross-correlation function between subsequent images. Tracking of diffusing atoms has been done manually on aligned image series. Only atoms with a clear identifiable random walk have been used in the quantification. Additionally, only areas that are at least 1 μm away from the nearest particle have been considered as the particles impact the surrounding concentration of impurities (see Supplementary Fig. 12). The distance between start and finish of those random walks was taken at the maximum time of observation for each atom individually. The observation time is mainly limited by drift (atoms leave the field of view), aberrations or atoms moving to an interface or surface. For the calculation of the coefficient of diffusion only in-plane movement was considered. The coefficient of diffusion was calculated for each atom individually and then all coefficients for one temperature were averaged for the final number. The observation of diffusion at grain boundaries was done after an initial annealing step to ensure that grain growth and boundary migration are not dominating the mechanism. STEM image simulations were performed using the open-source software Prismatic[37].

## Data availability

Relevant data supporting the key findings of this study are available within the article and the Supplementary Information file. All raw image files that support the findings of this study are available from the corresponding author upon request. Source data are provided with this paper.

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

## Acknowledgements
The authors would like to acknowledge the technical support of Gerhard Bürki.

## Author contributions
P.S. conceived and conducted the experiments, analysed the data, and wrote the manuscript. A. S. and L.V. helped analyse the data and write the manuscript. L.P., E.H. and A.S. synthesised the samples. J.M. and X.M. supervised the project.

## Competing interests
The authors declare no competing interests.
