## [Peer Review File · Nature Communications]

REVIEWER COMMENTS

Reviewer #1 (Remarks to the Author):

The manuscript "Atomic Scale Volume and Grain Boundary Diffusion elucidated by in situ TEM" by Schweizer et al. reports an innovative experimental approach to employ scanning TEM to study the volume diffusion behavior of W, a substitutional element in the fcc lattice of Cu. The single atoms of W can be visualized by STEM-HAADF contrast and the random motion of the W atoms in the lattice are followed in a temperature regime selected to have sufficient mobility and fast enough recording to track the motion. The authors select a [100] zone axis allowing to resolve jumps with $\frac{1}{2} \langle 110 \rangle$ distance, i.e. between nearest neighbor positions. Jumps out of a (100) plane into the next layer above or below can be still tracked within the depth of focus and identified by the jump distance. Thus, the random walk can be tracked.

The authors also performed beam-on and beam-off experiments to resolve possible impact of the electron beam on atomic motion. In addition, image simulations were performed to demonstrate agreement with the experimental HAADF image intensity. I also like the sections where the twin boundary and the random grain boundary are studied. The paper is highly innovative and the authors have carefully analyzed the data. I have only very few questions for clarification:

Can the authors comment at what depth the W atoms are located and how they are discriminated from possible W atoms at the surface? In the same context: can you exclude that in some cases you are close to the surface and the W atom jumps possibly between surface and volume - very much alike what you show for the twin boundary diffusion where the W atom always comes back to the GB?

I like the beam-on-off experiments, which gives credit to your results! Just for curiosity, why didn't you perform the experiments at 100 or 120 KV far below the knock-on threshold; also 2 different beam currents would have been interesting to check for possible impact on jump frequencies. Can you comment? (I am not asking to perform the experiments, but would like to hear why you didn't go for it or if you tried and what came out).

Reviewer #2 (Remarks to the Author):

P. Schweizer et al. have reported on atom diffusion in the volume and around the grain boundary using in-situ heating STEM imaging and quantitative analysis. So far, the diffusion experiments have been limited to tracing the isotope or radioactive atoms, which makes it difficult to examine the diffusion mechanism, although it can quantitatively measure the diffusion coefficient of dopants. However, as the author pointed out, it is usually difficult to separately measure the diffusion coefficients between the volume and the grain boundary (although there are several reports on the grain boundary diffusion using bicrystals). In this report, the authors use atomic-resolution electron microscopy and successfully track the dopant atoms by adequately selecting the observation regions, such as bulk and twin boundary, which makes it possible to measure the diffusion coefficient as a function of temperature separately. The manuscript is clear and well-written, and the topic should have broad interests. Therefore, the reviewer is pleased to recommend the publication of this manuscript after the consideration of several queries below.

1. In this manuscript, the experimental setup and analysis procedures are not well given, so please describe in detail in the Methods section—for example, the used electron-beam current, and acquisition time (or frame rate). The (estimated) concentration of W in copper should be noted, which may affect the experiments (probably dilute enough in this experiment).

2. In this experiment, the authors used STEM rather than TEM, and therefore the title may be better to use "in-situ STEM" rather than in-situ TEM.

3. In high temperatures, the authors can ignore the effects of the electron beam (knock-on). However, at lower temperatures, the knock-on process could be the main reason for W diffusion. If the authors measure the diffusion coefficient at room temperature, it should be given in Fig. 3.

4. For the estimation of the diffusion coefficient, how to calculate the following parameters: (i) Is

the out-of-plane diffusion corrected by the power of (3/2)? (ii) how to evaluate the diffusion distance (integrating over the time per frame or the distance between starting and ending points)?

5. In Fig. 1c, one can see the vague W atoms, which could be the surface diffusion rather than the volume diffusion. Therefore, it might be good to comment on this manuscript.

6. The diffusion of W happens under the lattice constraint of the copper, and the diffusion should be very different from an ideal gas. The reviewer does not understand the meaning of the discussion of diffusion speed. If there is no physical meaning, the authors should delete this discussion.

7. The segregation behavior is totally different from the meaning of complexion, as the author cited, which may not be suitable for this manuscript.

8. Grain boundaries have respective characters, so it will not be easy to understand from a few observations. In the present twin boundary case, it may simply formulate as the last equation. However, there are many atomic sites in general grain boundaries, and the interatomic distances vary. Therefore, the discussion on the GB formation energy may not be suitable.

Reviewer #3 (Remarks to the Author):

Addressing the challenge of investigating diffusion on an atomistic scale in bulk material is novel and timely.

Most studies so far have been focused on single atom surface diffusion. As such it is an essential contribution to the understanding of diffusion mechanisms especially at relatively low homologous temperatures, where a deviation from the high temperature Arrhenius type behavior has been observed in indirect studies. Secondly, the determination of twin boundary diffusion data is impressive.

The referee recommends publication of this paper after major revisions following the specific comments.

Specific Comments:

The materials and the arguments for the materials choice should be presented much earlier in the paper and also included into the captions of the figures.

Add a discussion about how the results of the authors fit in with results on surface diffusion?

If the W impurity atoms originate from the W nanoparticles in the Cu matrix there must be a radial concentration gradient away from the nanoparticle matrix interface. Was this gradient observed in the experiment and did it bias the random walk?

As a thin TEM lamella also contains two surfaces, why do they not contribute to the diffusion process. What would be the ratio between the diffusion constants between bulk and surface?

Typo on page 3, line 18: temperature unit missing

Please calculate the energy input by the electron beam for both cases aluminum and copper and explain quantitatively, why it can be ignored in copper and to what error this leads in terms of quantifying diffusion data. This error must be temperature dependent and thus affect the activation energy.

How can the results be generalised for other systems?

RESPONSE TO REVIEWERS' COMMENTS

We would like to thank all three reviewers for thoroughly reviewing our manuscript and raising excellent and insightful points. In response to these comments, we made several changes to the manuscript and added three new supplementary figures providing additional information and clarifications. Any changes to the manuscript and supplementary information are highlighted in green. A detailed response to the reviewers and their comments follows below.

Reviewer #1:

We thank the reviewer for kindly recommending our manuscript for publication, which they call “highly innovate”. We gladly address the reviewer’s questions point by point below.

1. *Can the authors comment at what depth the W atoms are located and how they are discriminated from possible W atoms at the surface?*

We assume that the atoms are randomly distributed throughout the volume due to the simultaneous co-deposition process which gives us a continuous flux of impurity atoms and matrix atoms. Unfortunately, the cross-sectional samples we prepared were not thin enough to see the atoms directly in high resolution, however, the co-deposited nanoparticles are distributed throughout the film (as shown in supplementary figure 1. This gives us confidence that the atoms are indeed dispersed throughout the film as well.

Beyond this general assessment of the deposition depth of impurity atoms, we can also distinguish between volume and surface atoms by different means: On the one hand volume atoms can be distinguished from surface atoms indirectly by their behavior at elevated temperatures: surface atoms move much faster and have different jump characteristics than volume diffusion. In fact, for surface atoms, we expect (and see) diffusion already at room temperature. At elevated temperatures the diffusion on the surface will be too fast to see the atoms at all (estimated displacements of micrometers per second at 350 °C, see also response to point 4 by reviewer 3) which would mean that any atoms observed at these temperatures must be contained within the volume or trapped at the interface to the substrate.

On the other hand, there is the possibility to differentiate between surface and volume by slight differences in the HAADF Signal for different depths of the impurity. This approach has been described previously¹ and relies on complementary image simulations. Indeed, performing such simulations we can at least qualitatively confirm that the contrast given by atoms looked at in this study fits better to atoms in the volume than on the surface. These results are now shown in supplementary figure 3.

To reflect this excellent point by the reviewer we have included an additional supplementary figure containing image simulations which are compared to the experiments and expanded the discussion surrounding surface diffusion.

2. *In the same context: can you exclude that in some cases you are close to the surface and the W atom jumps possibly between surface and volume - very much alike what you show for the twin boundary diffusion where the W atom always comes back to the GB?*

We would like to thank the reviewer for making this excellent point. Indeed, during observation we see Atoms seemingly vanish or appear during observation. This can be seen for instance in supplementary movie 1, where atoms seem to appear in the top left of the frame. We attribute this to atoms entering (or leaving) the volume from the surface. Transport over the surface is too fast to be seen at elevated temperatures which will give the impression that atoms appear and disappear out of nowhere. In response to this point we added this observation to the manuscript.

3. *I like the beam-on-off experiments, which gives credit to your results! Just for curiosity, why didn't you perform the experiments at 100 or 120 KV far below the knock-on threshold; also 2 different beam currents would have been interesting to check for possible impact on jump frequencies. Can you comment? (I am not asking to perform the experiments, but would like to hear why you didn't go for it or if you tried and what came out).*

We thank the reviewer for making this point. We agree that it would be great to repeat these experiments with a lower voltage in the future which would also potentially make other diffusion couples accessible (like Al as a matrix element). The reason we focused on 200 keV simply stems from the fact that we only had 200 and 80 keV available in our instrument at the time. We tried the experiment with 80 keV but simply could not get the necessary spatial resolution while at the same time having a high enough frame rate to observe diffusion. We did however try different dose rates at 200 keV. In total we varied the dose by a factor of 10 from around $1 \cdot 10^4$ to $1 \cdot 10^5$ e⁻/Å²/s. While there was no apparent difference for W in Cu, we saw an increase of atomic motion with W in Al. This fits well with our assumption that knock-on damage in Al causes enhanced diffusion. Due to this comment and comments by reviewers 2 and 3 we included more details about knock-on damage and its effects on aluminum and mentioned the dose rates in the method section.

Reviewer #2:

We would like to thank the reviewer for reading and recommending our manuscript. We are especially pleased that they find it “well written” and of “broad interest”. We gladly address the reviewer’s queries point by point below.

1. *In this manuscript, the experimental setup and analysis procedures are not well given, so please describe in detail in the Methods section—for example, the used electron-beam current, and acquisition time (or frame rate). The (estimated) concentration of W in copper should be noted, which may affect the experiments (probably dilute enough in this experiment).*

We thank the reviewer for suggesting to include more experimental details which we will gladly follow. The concentration of W in copper was calculated to be in the range of

around 150 ppm. In response to this comment we added appropriate parts in the methods section and included the impurity concentration in the manuscript and supplementary Figure 1.

2. *In this experiment, the authors used STEM rather than TEM, and therefore the title may be better to use “in-situ STEM” rather than in-situ TEM.*

We thank the reviewer for their valuable suggestion. Our initial thought of using ‘TEM’ was to address the whole technique which may be more familiar to a broader audience. However, we agree with the reviewer that STEM is more appropriate. We changed the title of the manuscript accordingly.

3. *In high temperatures, the authors can ignore the effects of the electron beam (knock-on). However, at lower temperatures, the knock-on process could be the main reason for W diffusion. If the authors measure the diffusion coefficient at room temperature, it should be given in Fig. 3.*

We thank the reviewer for this suggestion. At room temperature we are not able to measure any diffusion at all for W in Cu as one would expect (which is why it is not given in Fig.4). As mentioned in the manuscript the threshold for observation lies around 250 °C. Below that temperature diffusion still occurs but the timescales are just too long to observe it. Calculating the maximum energy transferred by the electron beam (8.2eV) and comparing it against the threshold displacement energy (19 eV for copper), we do not expect any knock-on effects for this system (see also reviewer 3, point 6).

However, for W in Al we do observe room temperature diffusion (as shown in supplementary figure 7), which we believe stems from knock-on effects. Indeed, calculating the energy transfer by the electron beam (19.5 eV) and comparing it against the threshold displacement energy (16 eV) we do see knock-on effects in this system that lead to room temperature diffusion. The coefficient of diffusion for W in Al at room temperature is in the range of 10^{-22} m²/s.

In response to this comment (and also a similar comment by reviewer 3) we decided to include the energies transferred by the electron beam as well as threshold displacement energies. Furthermore, we added a supplementary figure that shows the quantification of diffusion of W in Al to include the room temperature diffusivity.

4. *For the estimation of the diffusion coefficient, how to calculate the following parameters: (i) Is the out-of-plane diffusion corrected by the power of (3/2)? (ii) how to evaluate the diffusion distance (integrating over the time per frame or the distance between starting and ending points)?*

(i) The out-of-plane movement is not considered for the calculation of the diffusion coefficients. For an isotropic material, diffusion can be evaluated in different dimensions independently as $D_x=D_y=D_z$. Here we only look at diffusion in the sample plane (D_x and D_y) for our calculation and assume that diffusion in the z-direction has the same magnitude. We hope that future work will be able to address the out-of-plane movement more adequately.

(II) The diffusion distance is evaluated as demonstrated in Figure 2: for a given atom, the distance between start and end point is taken at the longest time the atom was observed. The time we can observe one atom is limited by factors such as drift, aberrations or when an atom moves towards a grain boundary. With this distance the atom has moved and the observed time we can calculate the diffusion coefficient. In response to this question, we expanded the methods section and the description of Figure 2 to make our process more understandable.

5. *In Fig. 1c, one can see the vague W atoms, which could be the surface diffusion rather than the volume diffusion. Therefore, it might be good to comment on this manuscript.*

We thank the reviewer for their keen observation. Indeed, looking closely at the bottom left of figure 1 we can see additional contrast potentially corresponding to W atoms which we did not highlight. We did consider different potential configurations for those atoms, one of which is an immobile surface position, which may occur for instance at the interface to the silicon nitride substrate. A mobile surface site is highly unlikely as surface diffusion at that temperature would immediately transport the atoms away at that temperature (see also discussion around point 4 of reviewer 3). Another explanation may be that of a grain boundary site, as there is an inclined grain boundary coming from the left. This can be seen in supplementary figure 3 where lattice planes originating from a different grain are visible on the left and overlap with the highlighted grain. At this point we are not sure what the explanation is, which is why we did not include these atoms in our calculations. However, we agree with the reviewer that it would be worth mentioning these atoms and their potential character in the manuscript.

To reflect this point, we decided to add this to the description of Figure 1c and supplementary Figure 4.

6. *The diffusion of W happens under the lattice constraint of the copper, and the diffusion should be very different from an ideal gas. The reviewer does not understand the meaning of the discussion of diffusion speed. If there is no physical meaning, the authors should delete this discussion.*

We thank the reviewer for asking for clarifications on this point. Maybe the term 'not directly physical meaningful', as we put it in the text is misleading. What we tried to do in this section is show that the distribution of apparent 'speeds' (speed here means the distance an atom travels in a specific time frame which varies from atom to atom) does follow a Maxwell-Boltzmann type distribution function just like in an ideal gas. Fitting the distribution gives us a single parameter (the equivalent mass). With this parameter the characteristics of the diffusion process can be described which we believe could be highly interesting for modelling of diffusion. Of course, the physical process of diffusion in a solid is very different from that of an ideal gas. However, we still think it is very valuable to draw parallels here. We of course agree with the reviewer that this part of the manuscript should be more precise. To address this point, we changed the paragraph to make it clearer what we mean.

7. *The segregation behavior is totally different from the meaning of complexion, as the author cited, which may not be suitable for this manuscript.*

We thank the reviewer for making this excellent point and agree that complexion is not the right word to describe the type of grain boundary segregation that we see. In response to this point we changed the wording from complexion to segregation in the manuscript and the SI.

- 8. Grain boundaries have respective characters, so it will not be easy to understand from a few observations. In the present twin boundary case, it may simply formulate as the last equation. However, there are many atomic sites in general grain boundaries, and the interatomic distances vary. Therefore, the discussion on the GB formation energy may not be suitable.*

We agree that grain boundaries can have very different characters depending on their configuration with many grain boundary sites to consider. The relation of the grain boundary energy to the diffusivity used in this manuscript is a strong simplification and may not be suitable for arbitrary grain boundaries. However, we believe - and the reviewer seems to agree- that it may be well applicable to twin boundaries. We follow the reviewer's assessment that the discussion around that point should mention that the aforementioned equation may not be applicable to arbitrary boundaries. In response we added a sentence to the discussion about grain boundary energies.

Reviewer#3

We would like to express our gratitude to the the reviewer for taking the time to review our manuscript which they describe as "novel" and parts of the data as "impressive". We thank them for recommending it for publication and gladly address their comments point by point below.

- 1. The materials and the arguments for the materials choice should be presented much earlier in the paper and also included into the captions of the figures.*

We thank the reviewer for suggesting this point. In response we noted the material choice earlier in the manuscript (on page 1 line 29) and also updated the caption of figure 1.

- 2. Add a discussion about how the results of the authors fit in with results on surface diffusion?*

We thank the reviewer for suggesting to add to the discussion by including context regarding surface diffusion. In response to this point and also to point 4 of the same reviewer we added to the discussion of surface diffusion effects.

- 3. If the W impurity atoms originate from the W nanoparticles in the Cu matrix there must be a radial concentration gradient away from the nanoparticle matrix interface. Was this gradient observed in the experiment and did it bias the random walk?*

We thank the reviewer for their great question. While there are some nanoparticles being deposited in the films, the parameters are tuned in such a way that most atoms do not agglomerate into particles. For our diffusion experiments we focused only on areas that are far away from these particles to exclude any effects that they may have. The reviewer is correct in assuming that the particles can have an effect on the radial concentration of atoms in the close

vicinity of them. We found that, depending on the size of the particles they could act either as sinks or as sources of atoms, with smaller particles dissolving and bigger particles capturing more impurities. We found that the area of influence is rather small, in the range of 10-20 nm. To reflect this discussion, we added a supplementary figure showing nanoparticles and surrounding atoms during heating and a corresponding section in the method section.

4. As a thin TEM lamella also contains two surfaces, why do they not contribute to the diffusion process. What would be the ratio between the diffusion constants between bulk and surface?

We thank the reviewer for making this point. Indeed, surface diffusion does happen on our specimen and contributes to the overall diffusion process. While one surface is capped by silicon nitride where only interface diffusion may take place, the other surface is indeed open for surface diffusion. Surface diffusion is generally much faster than volume diffusion, which is why (as stated in the manuscript) we already see it at room temperature. At higher temperatures surface diffusion becomes so fast that we cannot observe it anymore directly on the atomic scale. To give some numbers: at room temperature the surface self-diffusion of tungsten would lead to mean displacements of a few angstroms per s, which can be seen with HRSTEM (as shown in see supplementary Figure 6.). Increasing the temperature to 350°C the surface self-diffusion coefficient of tungsten is as high as 10^{-11} m²/s which would indicate mean displacements of micrometers per second. This is much faster than volume diffusion for which (in our system) the coefficient of diffusion is in the range of 10^{-21} m²/s, close to 10 orders of magnitude lower.

In general, surface diffusion leads to phenomena such as dewetting, which we do observe in our samples as well especially at temperatures above 400 °C. However, for our analysis we focused only on clearly resolvable volume diffusion events that were recorded after a steady state in surface diffusion events has been achieved. In response to this point (and point 2 by the same reviewer) we included some numbers for surface diffusion and extended the discussion around that topic.

5. Typo on page 3, line 18: temperature unit missing.

We thank the reviewer for carefully reading our manuscript and catching this typo. We added the unit accordingly.

6. Please calculate the energy input by the electron beam for both cases aluminum and copper and explain quantitatively, why it can be ignored in copper and to what error this leads in terms of quantifying diffusion data. This error must be temperature dependent and thus affect the activation energy.

We thank the reviewer for this valuable suggestion. Calculating the maximum possible energy transferred by a single electron to copper and aluminum we obtain about 8.2 eV and 19.5 eV. For aluminum this is above the threshold displacement energy of 16 eV while for copper it is below the threshold displacement energy (19 eV)². This means that aluminum atoms can directly be displaced from the lattice whereas for copper this is not possible. Direct displacement from the lattice means that additional vacancies are generated which increase diffusivity. This effect can be seen in aluminum but is absent in copper. For aluminum we see that the activation energy is strongly affected by knock-on, giving us values as low as 0.3 eV.

Most energy transferred by the electron beam that does not lead to knock-on displacements will still lead to sample heating. This heating effect can be estimated as follows³:

$$\Delta T \approx I \cdot E_m \ln\left(\frac{R_0}{R}\right) / (2\pi\kappa\lambda)$$

With our parameters we find a heating effect of about 0.03 K (Parameters 200 keV, 200 pA, probe size $R=0.1$ nm, heat conductivity of copper $\kappa=401$ W/(m*k), Distance to grid $R_0= 30$ μm , average energy transfer per electron $E=8.2$ eV (to be on the safe side we assume that the maximum energy is transferred in each collision) and inelastic mean free path: 100 nm⁴). This value is well in line with other reported values for good heat conductors. The heating effect is below the uncertainty of the temperature given by the heating equipment and does therefore not significantly affect the uncertainty of the measurement or the activation energy. If the experiments are repeated with a worse heat conductor, beam heating effects need to be taken into account.

To reflect this point, we added to the discussion about knock on displacement by providing the calculated values. In addition, we describe the beam heating effect in the method section. Finally, we included a supplementary figure with the quantification of Al in the SI, showing a strong effect of the electron beam.

7. How can the results be generalised for other systems?

We thank the reviewer for this great question! We believe that the diffusion dynamics shown in this paper can be generalized to other FCC metals such as Nickel or Gold when they contain dilute metal impurities. In those systems all the mechanistic observations made in our manuscript should be transferable albeit with different numbers. There may be limited applicability to intermetallic systems (e.g. L₁₂ ordered systems) which still have an FCC lattice but due to chemical ordering could have anisotropic diffusion. We hope that this work will serve as only the starting point and will kickstart studies on diffusion in other systems as well.

In response to this point we added a sentence in the summary and conclusion section of the manuscript.

References:

1. Ishikawa, R. *et al.* Direct Observation of Dopant Atom Diffusion in a Bulk Semiconductor Crystal Enhanced by a Large Size Mismatch. *Phys. Rev. Lett.* **113**, 155501 (2014).
2. Makin, M. J. Electron displacement damage in copper and aluminium in a high voltage electron microscope. *The Philosophical Magazine: A Journal of Theoretical Experimental and Applied Physics* **18**, 637–653 (1968).
3. Egerton, R. Radiation Damage and Nanofabrication in TEM and STEM. *Microscopy Today* **29**, 56–59 (2021).

4. Iakoubovskii, K., Mitsuishi, K., Nakayama, Y. & Furuya, K. Mean free path of inelastic electron scattering in elemental solids and oxides using transmission electron microscopy: Atomic number dependent oscillatory behavior. *Phys. Rev. B* **77**, 104102 (2008).

REVIEWERS' COMMENTS

Reviewer #1 (Remarks to the Author):

The authors have addressed all points of concern in the revised version. I suggest publication of this highly interesting and novel work with real-time STEM measurements of diffusion at atomic scale.

Reviewer #2 (Remarks to the Author):

Dear authors,

I thank the authors for their detailed responses and proper revisions. The manuscript has been well improved, and I am pleased to recommend the publication of this manuscript.

Reviewer #3 (Remarks to the Author):

The authors have adequately addressed my questions and comments and I recommend this paper for publication.

RESPONSE TO REVIEWERS' COMMENTS

We thank the reviewers for once again reviewing our manuscript and evaluating our responses to their initial comments. We are delighted that the reviewers find the changes that we made to the manuscript during revision fully satisfactory and that all of them recommend our manuscript for publication. Therefore, no further changes were made to the manuscript other than for editorial purposes.